# Ideas and Perspectives:

# Climate-Relevant Marine Biologically-Driven Mechanisms in Earth System Models

Inga Hense[1], Irene Stemmler[2], and Sebastian Sonntag[2]

[1]IHF, Center for Earth System Research and Sustainability, University of Hamburg, Germany

[2]Max Planck Institute for Meteorology, Bundesstrasse 53, 20146 Hamburg, Germany

*Correspondence to:* Inga Hense (inga.hense@uni-hamburg.de)

**Abstract.** The current generation of marine biogeochemical modules in Earth System Models (ESMs) considers mainly the effect of marine biota on the carbon cycle. We propose to also implement other biologically-driven mechanisms in ESMs so that more climate-relevant feedbacks are captured. We classify these mechanisms in three categories according to their functional role in the Earth system: (1) "biogeochemical pumps" which affect the carbon cycling, (2) "biological gas and particle shuttles" which affect the atmospheric composition and (3) "biogeophysical mechanisms" which affect the thermal, optical, and mechanical properties of the ocean. To resolve mechanisms from all three classes, we find it sufficient to include five functional groups: bulk phyto- and zooplankton, calcifiers as well as coastal gas and surface mat producers. We strongly suggest to account for a larger *mechanism diversity* in ESMs in future to improve the quality of climate projections.

## 1 Introduction

This "Ideas and Perspectives" paper deals with the role of marine biota in the climate system and the way this role can be adequately captured in the marine ecosystem components of Earth System Models for climate research.

The representation of the marine ecosystem in Earth system models (ESMs) used for climate projections has been significantly refined in recent years. Plankton, for example, has been split into functional groups, and physiological details, such as light or nutrient acclimation, have been added (e.g. Vichi et al., 2011; Aumont and Bopp, 2006; Aumont et al., 2015). Most of these modifications had been motivated by studies regarding the impact of climate change on marine ecosystems, or improving the representation of biogeochemical cycles, specifically the carbon cycle. Little attention, however, has been paid to other biologically mediated climate-relevant mechanisms, which we define as combinations of processes that lead to climate

feedbacks. Here, we will present a framework to classify these biological-chemical-physical mechanisms and the functional groups that are necessary to describe them.

Many of today's marine biogeochemical models used in ESMs for climate projections include several phyto- and zooplankton functional groups; in some cases even variations in element or chlorophyll content of organic matter are allowed. (see Laufkötter et al., 2015, for an overview). Apart from discussions about the appropriate degree of complexity in biogeochemical models (see Anderson (2005), Flynn (2006) and Le Quéré (2006)), even the most complex models "only" refine the representation of the marine carbon cycle. The climate-carbon cycle feedback, however, is just one of several feedback loops in which marine biota interacts with other components of the climate system.

These additional links are or may become important for the evolution of the climate system and should be implemented in ESMs. Thus, instead of adding more details to better represent just one mechanism, we should account for a "mechanism diversity". This way, the consequences of an altered functioning of the marine ecosystem with climate change will feed back on the climate system in multiple ways.

To adequately account for the proposed mechanism diversity, the first task is to come up with a list of relevant mechanisms. We define biologically-driven mechanisms to be *climate-relevant* on timescales of contemporary climate change if they lead to a change in global energy (heat) content and distribution. These are, with decreasing levels of directness: (i) mechanisms with an immediate impact on the planetary albedo and/or sea surface temperature, (ii) mechanisms which change the content and distribution of greenhouse gases or ocean's turbulent viscosity, and (iii) mechanisms which change for instance the ocean's nutrient inventory with potential consequences for the marine carbon cycle and thus atmospheric greenhouse gas concentrations. Because the climate relevance of mechanisms on the third level is difficult to evaluate, we will limit this discussion to those of the first and second level. Even for these, quantitative estimates about the impact on the global energy budget are not available in all cases. Often, however, useful semi-quantitative evaluations, for example on ocean circulation patterns, exist and we will use them instead.

We will present a general framework that illustrates the links between the marine biota, the mechanisms and the larger feedback loops in the climate system in a systematic way. Within this framework, individual processes as part of the mechanisms will be described only briefly, and only, if they are indispensable for a basic understanding. Our list of processes cannot be complete, yet all mechanisms will be presented at a comparable level of abstraction. We believe that the framework will prove a useful basis for classification, even if additional biological climate-relevant mechanisms are discovered.

## 2    What is needed: A classification of biologically driven mechanisms

We adopt the idea to split the marine biota into different groups, but in contrast to previous approaches, we classify them according to their functional role in the *climate system*. The functions these organism groups carry are drivers of climate-relevant mechanisms.

This leads us to three classes of mechanisms (M1-M3) that generate climate feedbacks (see Fig. 1). For each class we briefly explain the main mechanisms, present the key organisms involved, and highlight the

climate relevance. Finally, we describe the functional groups needed to represent this mechanism in ESMs (Table 1).

## M1 - biogeochemical pumps

The first class of mechanisms comprises the marine part of the carbon cycle, including the *organic carbon pump*, *the microbial carbon pump*, and the *alkalinity pump*.

The *organic carbon pump* includes the processes related to the uptake of carbon dioxide in the upper ocean and the sinking of organically bound carbon to deeper waters. Three main organism groups are involved – phytoplankton, zooplankton, and bacteria. Phytoplankton drives the carbon cycle, because inorganic carbon is transferred to organic carbon via photosynthesis and zooplankton decisively contributes to carbon export to the deeper ocean via fecal pellet production. Bacteria decompose the organic matter while it is sinking down and thereby determine the efficiency of the organic carbon pump. The climate relevance of the organic carbon pump has been evaluated in several model studies: Rough estimates suggest that atmospheric $CO_2$ levels would rise by approximately 200 ppmv after a complete shutdown of the organic carbon pump (Volk and Hoffert, 1985; Broecker and Peng, 1986). As part of the climate-carbon cycle feedback (Friedlingstein et al., 2006), this mechanism is well known and regarded as the most important marine biologically-driven mechanism. To capture the organic carbon pump in ESMs, two functional groups are, in principle, sufficient – a bulk phytoplankton and a bulk zooplankton group to describe the transformation process from inorganic to organic matter and sinking of the latter. All additional functional groups that are needed for other mechanisms, however, will also contribute (see Table 1). Bacteria do not need to be explicitly included as a key group to adequately represent the organic carbon pump, because bacterial decomposition can be assumed to be roughly proportional to the available organic matter.

The *microbial carbon pump* describes the pathway from more easily degradable to refractory organic carbon by microbes (e.g. Jiao et al., 2010). These organisms transform dissolved or particulate organic carbon into compounds that are resistant towards degradation and are therefore stored for thousands of years. The refractory organic carbon pool is large and comparable to the atmospheric $CO_2$ reservoir (Hansell et al., 2009), but it will have little impact on the climate system on time scales of several hundreds of years, unless an imbalance between sources and sinks evolves. Although it has been speculated that such changes may occur under ocean acidification and eutrophication (Jiao et al., 2014), there is insufficient knowledge to account for the microbial carbon pump and the corresponding functional groups in ESMs. In addition, no evaluation of the relevance of this pump with respect to contemporary climate change yet exists.

The *alkalinity pump* is another essential part of the marine carbon cycle, because this mechanism alters the carbonate chemistry in the ocean. Organisms that affect the carbonate equilibrium are calcifying species, forming calcite or aragonite shells. They occur in the open ocean (e.g. coccolithophores) as well as in shallow regions (e.g. corals) where they "consume" alkalinity and release $CO_2$ during the calcification process, causing a decrease in alkalinity. Since alkalinity is the capacity of the ocean to buffer acids and sets the limit how much $CO_2$ can be stored, changes in alkalinity have consequences for the $CO_2$-storage. While the quantitative impact of the alkalinity pump on climate is currently unclear, its role via the "calcification

feedback" on atmospheric $CO_2$-concentrations is assumed to be large (Zhang and Cao, 2016). Among the calcifiers, coccolithophores are the most important group (see e.g. Rost and Riebesell, 2004) and mainly responsible for the vertical gradient in alkalinity. When coccolithophores die, they sink down to the deeper part of the ocean, where the calcareous shells dissolve and the alkalinity increases. Other calcifying or-
ganism groups have been shown to be regionally important (see e.g. Baumann et al., 2004; Kleypas et al., 2006) or are assumed to be relevant for aragonite (Gangstø et al., 2008) but presumably only marginally for climate dynamics. To represent the alkalinity pump in ESMs, calcifiers need to be included to generate the vertical alkalinity gradient and to adequately resolve the carbonate chemistry. From a climate perspective, the gain from representing calcifiers by more than one key group might be relatively small unless regional
ESMs are applied. With one additional key group, the calcifiers, represented by coccolithophores, the basic features of the alkalinity pump would be captured.

**M2 - biological gas and particle shuttles**

The second class of mechanisms, the "*biological gas and particle shuttles*", addresses the impact of the marine biosphere on the atmosphere due to emission of gases and particles. These substances belong to the
group of "short-lived climate-relevant air contaminants" (SCC), a subset of short-lived health- and climate-relevant air contaminants (SHCC), sensu Pöschl and Shiraiwa (2015). They may act as aerosols, influencing cloud formation. They may also affect the atmospheric chemistry or influence the thermodynamics as greenhouse gases.

Particulate SCCs of marine biogenic origin directly affecting cloud formation are called "marine bio-
genic primary aerosols". These include entire organisms, like phytoplankton cells or organisms' remnants, or "exudates", which are substances secreted by organisms (e.g. Knopf et al., 2011; Burrows et al., 2013; Wilson et al., 2015). Although the research area of marine biogenic aerosols is relatively new, recent studies suggest that at least on a regional scale, ocean biota strongly influences the concentrations of cloud droplets with significant consequences for the reflected shortwave radiation (McCoy et al., 2015). Thus,
ocean biota as a source for primary aerosols can directly contribute to the cloud-albedo feedback. As a first approximation, no additional functional group needs to be added in ESMs; a fraction of those organisms in the surface layer that are implemented in ESMs anyway may serve as a source for primary aerosols.

Gaseous SCCs may be involved in aerosol formation or participate in ozone reactions. The most important gaseous SCCs produced by marine organisms are dimethyl sulfide (DMS) and short-lived halocarbons.
For both of these it is meaningful to distinguish open and coastal ocean sources since their efficiency in gas release is highly dissimilar and different organism groups are involved. DMS (or its precursor) is produced by "open ocean" (coccolithophores) and "coastal" phytoplankton (*Phaeocystis*) groups (e.g. Barnard et al., 1984; Malin et al., 1993). Zooplankton and bacteria are involved (Reisch et al., 2011) and similar to the organic carbon pump, especially bacteria determine the efficiency of the DMS shuttle to a large extent.
Short-lived bromine halocarbons are associated with "open ocean" phytoplankton and "coastal" macroalgae (e.g. Moore et al., 1996; Nightingale et al., 1995; Carpenter and Liss, 2000).

Dimethyl sulfide (DMS) is a precursor of sulfate aerosols and involved in the cloud-albedo feedback (e.g. Charlson et al., 1987; Ayers and Cainey, 2008) although its climate relevance is still under discussion (e.g. Quinn and Bates, 2011). Local effects on shortwave radiation of DMS emission by a phytoplankton bloom can induce cooling up to 15 W m$^{-2}$ at the top of the atmosphere; such a high value is usually associated with heavily air-polluted regions (Meskhidze and Nenes, 2006). The global direct radiative effect of DMS has been estimated to be -0.23 W m$^{-2}$, the indirect as -0.76 W m$^{-2}$. The contribution of primary producers via DMS production to sources of natural aerosols is therefore larger than those from sea salt or volcanoes for example (Rap et al., 2013).

Short-lived halocarbons, particularly brominated substances are important SCCs, because they destroy ozone and thereby significantly change the radiative forcing (Sturges et al., 2000; Saiz-Lopez et al., 2012; Laube et al., 2008). The radiative effect is estimated to be about -0.2 W m$^{-2}$ and thus larger than the one by the widely known anthropogenically produced long-lived halocarbons such as CFCs (Hossaini et al., 2015).

For both DMS and short-lived halocarbons, it is crucial to correctly represent the spatial patterns of marine primary production and corresponding SCCs (e.g. Stemmler et al., 2015, for halocarbons). To capture the gradient between coastal and open ocean, an additional model compartment, the "coastal gas producers", has to be included in ESMs. A relatively easy way to describe them in the model is by allowing the sediment or deepest model layer being an additional nutrient pool and by taking into account relatively high emissions per unit biomass. Even if different types of organisms are involved in the coastal production of DMS and short-lived halocarbons, one functional group is sufficient, because coastal patterns of the two SCCs do not differ clearly. The group of open ocean organisms can be represented either by coccolithophores in case of DMS, or a "bulk phytoplankton" group in case of halocarbons (although parametrizations are necessary, because only part of the entire bulk phytoplankton produces halocarbons). Just like for the organic carbon pump, bacteria do not need to be explicitly considered for the DMS shuttle; zooplankton, the other group involved in DMS release, is included anyway, because of its role in the organic carbon pump.

Last but not least, there are a number of greenhouse gases of marine biogenic origin, notably $CO_2$. This gas is respired by all organisms and is more or less automatically captured in ESMs through the loss rate of all functional groups. In addition to $CO_2$, another important long-lived greenhouse gas is $N_2O$, which has a global warming potential of a 100 year time horizon that is approximately 300 times higher compared to $CO_2$ (Myhre et al., 2013). About 20% of the global production of $N_2O$ is of marine origin (Denman et al., 2007), mediated by microbes. $N_2O$ is released mainly during denitrification, the oxidation of ammonium to nitrate under oxic conditions, and to a lower extent during denitrification, the reduction from nitrate to dinitrogen gas under anoxic conditions (Freing et al., 2012). Two organism groups, bacteria and archaea, are involved in these transformation processes. So far, our knowledge regarding the spatial variations in the occurrence of the organisms involved and the respective rates is too fragmented to explicitly describe them in models. Instead, these bacterial transformation processes can be implicitly considered in the same way

as done for other mechanisms (by choosing turnover rates that are proportional to the available resources); thus no further model compartment is necessary.

Marine sources of other biogenic greenhouse gases like $CH_4$ are mainly related to marine microorganisms (e.g. Valentine, 2011). To the best of our knowledge, the effect of these greenhouse gases such as $CH_4$ on the climate system may be considered negligible, because the marine sources are small compared to the terrestrial or anthropogenic ones. Thus it is currently not justifiable to add more model compartments.

## M3 - biogeophysical mechanisms

The third class of biologically-driven climate-relevant mechanisms includes all *biogeophysical mechanisms*. These mechanisms comprise changes of thermal, optical and mechanical properties of the ocean, predominantly caused by phytoplankton species. Among them, positively buoyant cyanobacteria are particularly important, because they can produce surface mats of up to several millions of square kilometers (e.g. Capone et al., 1998). Such surface mats significantly change light absorption impacting the surface mixed layer heat balance (e.g. Sathyendranath et al., 1991; Kahru et al., 1993). In addition, they increase the albedo (e.g. Kahru et al., 1993), alter the turbulent viscosity, and reduce the vertical mixing (e.g. Jöhnk et al., 2008). Surface mats may also reduce the air-sea gas exchange, if we assume similar effects as for surface microlayers (e.g. Liss and Duce, 2005).

The climate impact of the light absorption mechanism has been only assessed for *neutrally buoyant* phytoplankton groups so far. Their impact alone, however, is significant: Pronounced effects on oceanic and atmospheric temperature, circulation patterns, cloudiness, humidity, precipitation and evaporation, as well as sea ice cover (Patara et al., 2012) and ENSO dynamics (e.g. Jochum et al., 2010) has been shown to be influenced through light absorption. The strong response triggered by this mechanism results from multiple feedback loops that involve different Earth system components.

Rough estimates indicate that changes of albedo through phytoplankton, specifically coccolithophores, can result in a cooling by roughly 0.2 W m$^{-2}$ globally (Tyrrell et al., 1999). A more sophisticated evaluation, however, points towards a negligible impact on the albedo, at least on the basin-scale (Gondwe et al., 2001). In any case there is a direct link to the albedo-temperature feedback (Watson and Lovelock, 1983).

Unfortunately, the climate effect through biotic induced changes of ocean's turbulent viscosity has not been addressed yet. Idealized model studies, however, suggest that biologically induced increase or decrease of turbulent viscosity by surface mats can affect ocean circulation patterns on a basin-scale (Sonntag, 2013).

To account for biogeophysical aspects in ESMs one additional key group, "surface mat producers" is needed. Cyanobacteria are a good candidate to represent this group. They possess the trait "positive buoyancy" which is not shared with other phytoplankton. Clearly, all other groups of marine primary producers that are explicitly described in ESMs have an impact on light absorption, too, but by distinguishing neutrally or negatively from positively buoyant phytoplankton a more realistic representation of the light absorption feedback will be achieved.

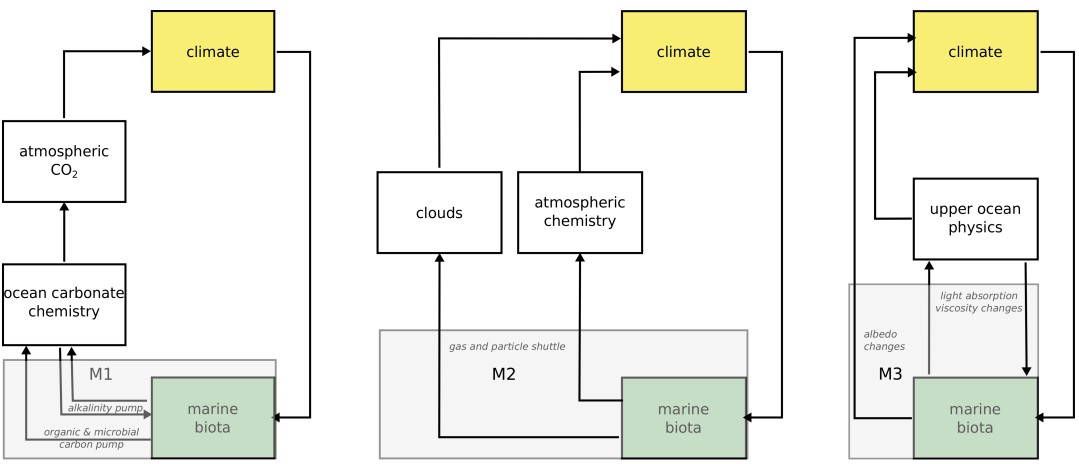

**Figure 1.** Major global climate feedback loops, based on the three classes of mechanisms (light grey-shaded boxes), driven by marine biota (green-shaded boxes). Only links originating from the marine biota are shown; additional inter- and cross-links between the different boxes are omitted for a clarity. a) the three mechanisms (the organic and the microbial carbon pump, and the alkalinity pump) affect the $CO_2$ inventory in the ocean, which in turn leads to changes in atmospheric $CO_2$ and thus in climate. An altered global climate influences the marine biota (through e.g. changes in SST, near surface stratification and circulation patterns), closing the marine part of the climate-carbon cycle feedback loop which also includes the $CO_2$-calcification feedback. b) the gas and particle shuttle alter cloud formation rates and distribution as well as atmospheric chemistry. There is a complex interplay between different atmospheric components that ultimately lead to climate change, again with consequences for the marine biota. A number of atmospheric feedbacks (e.g. the cloud-albedo feedback, the longwave radiation feedback, the chemistry feedbacks) are involved in this loop. Note that the influence of marine biota on *local* cloud cover is not illustrated here. c) two biogeophysical mechanisms (based on light absorption and turbulent viscosity changes) directly affect the upper ocean physics such as heat distribution and circulation and hence the biota. The third one (albedo changes) has a direct effect on the planetary radiation budget which influences in turn the marine biota.

| organism groups | M1 | M2 | M3 |
|---|---|---|---|
| bulk phytoplankton | ✓ | ✓ | ✓ |
| bulk zooplankton | ✓ | (✓) | - |
| calcifier | ✓ | ✓ | ✓ |
| coastal gas producer | ✓ | ✓ | ✓ |
| surface mat producer | ✓ | ✓ | ✓ |

**Table 1.** Organism groups that drive climate mechanisms 1: biogeochemical pumps; 2: gas and particle shuttle and 3: biogeophysical mechanisms. Note that zooplankton is partly involved in the production of DMS due to grazing and thus checked in parenthesis.

To summarize, including the above mentioned 5 functional groups (Table 1) will meet the requirements for an adequate representation of biologically-driven mechanisms in ESMs.

## 3 What changes may occur in future: Sensitivity of marine biota to climate stressors

The marine biota itself, as well as the strength of the individual mechanisms, may evolve under climate change due to changes in the three climate stressors – temperature, pH and oxygen. As a rule of thumb, higher temperatures increase the metabolic rates of organisms. Lower pH may increase growth of non-calcifiers and decrease that of calcifiers (e.g. Raven, 2011). Low oxygen concentrations will particularly

impact higher trophic levels and microbial processes. In principle, however, the response will be species-specific.

Among phytoplankton, cyanobacteria are assumed to strongly benefit from climate change, and thus they are expected to become more abundant in future (e.g. O'Neil et al., 2012; Hense et al., 2013). In particular, a moderate rise of sea surface temperature (Fu et al., 2014), as well as a decrease in pH, will favor their growth conditions (Hutchins et al., 2007). More cyanobacteria will intensify the biogeophysical feedback mechanisms and possibly the particle shuttle. The response of other phytoplankton to pH is not well understood. While ocean acidification may significantly affect calcifiers and the calcification rate, the response is not uniform (see e.g. Kleypas et al., 2006) and genetic adaptation (Lohbeck et al., 2012) might outweigh the negative consequences of a decreasing pH. Ocean acidification (but also increasing temperature) may directly affect DMS-producing organisms and thus outgassing of DMS: Depending on the grazing pressure, DMS production seems to be either enhanced (Kim et al., 2010) or reduced (Archer et al., 2013). A strong response of the climate system to reduced DMS-production on the radiative forcing has been proposed (Six et al., 2013).

Sensitivity to climate stressors has been also described for many microbial organisms. One example are nitrifiers. For lower pH, nitrification and therefore $N_2O$ production is strongly reduced (Beman et al., 2011). It is expected nevertheless that the production of $N_2O$ will increase in future (Naqvi et al., 2010) due to the expansion of oxygen minimum zones (Stramma et al., 2008), taking into account that the highest $N_2O$ production usually occurs at the anoxic-oxic interface. Another example of microbes that seem to benefit from climate change are those involved in aerobic decomposition of organic matter. A rise in temperature and a drop in pH stimulates bacterial turnover rates (Pomeroy and Deibel, 1986; Piontek et al., 2010). With enhanced remineralization, the efficiency of the organic carbon pump will be reduced, altering the ocean's carbon uptake capacity (Segschneider and Bendtsen, 2013). On the other hand, bacterial decomposition rates may be affected through a decrease in oxygen concentrations with an expansion of oxygen minimum zones (Stramma et al., 2008). It is still unclear, however, whether low oxygen concentrations will impair bacterial degradation of organic matter or not (see, e.g. Kristensen et al., 1995; Devol and Hartnett, 2001).

In addition to the immediate effect of climate stressors on ocean biota, we expect significant alterations in the environment with potentially large long-term consequences on the organisms and biologically-driven mechanisms. For example, the organic carbon pump will likely to be altered by changes in stratification (Steinacher et al., 2009), the ocean's molecular viscosity (Taucher et al., 2014), and plankton community composition (see, e.g. Laufkötter et al., 2016). The alkalinity pump may be affected by changes in freshwater input or evaporation (see e.g. Steinacher et al., 2009; Jiang et al., 2014). Overall, it is quite certain that the relative abundance of some phytoplankton organisms will change, as a result of their response to climate stressors and altered environmental conditions. Such a shift in community composition will affect the strength of all three classes of mechanisms, and with that their relative importance within the climate system.

## 4 What is currently done: The state of the art

Today's ESMs represent the first class of biologically-driven climate-relevant mechanisms, the biogeo-chemical pumps, (in particular the organic carbon pump) reasonably well (Table 2). Most of these models explicitly consider phyto- and zooplankton which are described in such a way that the model results give reasonable values for export production (see e.g. Ilyina et al., 2013; Palmer and Totterdell, 2001). The carbonate chemistry is also relatively well represented, even though calcifiers are not explicitly included but parametrized by assuming that they constitute a certain proportion of bulk phytoplankton.

The second class of mechanisms, which affect the atmospheric composition, has received less attention. Some ESMs do consider DMS/$N_2O$ (Table 2) and their results suggest significant changes in the production with consequences for the climate system in future (e.g. Six et al., 2013; Martinez-Rey et al., 2015). Other marine biologically produced SCCs (except $CO_2$) and aerosols are usually not included; but there is a number of recent modeling activities in which the pertinent processes have been implemented, and the climate impact of these substances has been partially evaluated (e.g. Kirkevåg et al., 2013; Stemmler et al., 2014, 2015; Hossaini et al., 2015). The largest deficiency of ESMs in this respect is that primary production is still not sufficiently well represented, in particular in coastal regions (e.g. Schneider et al., 2008; Anav et al., 2013). Even though the respective ESMs as well as global marine biogeochemical models have become more and more complex in recent years (see e.g. Aumont et al., 2003; Le Quéré et al., 2005; Dunne et al., 2013; Buitenhuis et al., 2013), the situation has only marginally improved. Not surprisingly, models generally fail to simulate SCC concentrations and air-sea fluxes on the shelf (see e.g. Halloran et al., 2010; Stemmler et al., 2015); much could be gained if coastal primary production is captured more realistically.

Finally, the third class of mechanisms, the marine biogeophysical mechanisms, are hardly addressed in today's ESMs; so far, only half of them include the light absorption mechanism involved in the feedback between the biota and temperature (Table 2). Recent studies with neutrally or negatively buoyant phytoplankton indicate that consequences for the upper ocean heat balance and the climate system are substantial (e.g. Patara et al., 2012; Lengaigne et al., 2009). Thus, the effects might be even stronger if positively buoyant organisms are added; whether organisms stay at the surface or whether they are homogeneously distributed in the surface mixed layer makes a big difference for the upper ocean heat budget (e.g. Sonntag and Hense, 2011). None of today's ESMs or coupled global biogeochemical ocean circulation models account for other biogeophysical effects, i.e. changes in albedo and turbulent viscosity.

| ESMs | MBMs | mechanisms | | | functional groups | | | | | |
|---|---|---|---|---|---|---|---|---|---|---|
| | | 1 | 2 | 3 | P | Z | C | cG | S | B |
| BCC_CSM1.1 | OCMIP | $\checkmark$ bcp | – | – | – | – | – | – | – | – |
| CanESM1 | CMOC | $\checkmark$ | – | – | $\checkmark$ 1 | $\checkmark$ 1 | – | – | – | – |
| CESM1 | BEC | $\checkmark$ | – | $\checkmark$ LA−nbp | $\checkmark$ 3 | $\checkmark$ 1 | ($\checkmark$) | – | – | – |
| ESM2M/ESM2G | TOPAZ2 | $\checkmark$ | – | $\checkmark$ LA−nbp | $\checkmark$ 3 | ($\checkmark$) | ($\checkmark$) | – | – | – |
| HadGEM2-ES | Diat-HadOCC | $\checkmark$ | $\checkmark$ DMS | – | $\checkmark$ 2 | $\checkmark$ 1 | ($\checkmark$) | – | – | – |
| CMCC-CESM | PELAGOS | $\checkmark$ bcp | – | $\checkmark$ LA−nbp | $\checkmark$ 3 | $\checkmark$ 3 | – | – | – | $\checkmark$ 1 |
| IPSL-CM5A | PISCES | $\checkmark$ | ($\checkmark$ DMS,N$_2$O) | $\checkmark$ LA−nbp | $\checkmark$ 2 | $\checkmark$ 2 | ($\checkmark$) | – | – | – |
| MIROC-ESM | NPZD | $\checkmark$ bcp | – | – | $\checkmark$ 1 | $\checkmark$ 1 | – | – | – | – |
| MPI-ESM | HAMOCC | $\checkmark$ | $\checkmark$ DMS | – | $\checkmark$ 1 | $\checkmark$ 1 | ($\checkmark$) | – | – | – |

**Table 2.** Different marine biosphere modules (MBMs) in Earth System Models (ESMs) that participated in CMIP5 (Arora et al., 2013; Laufkötter et al., 2015): OCMIP: Wu et al. (2013); CMOC: Christian et al. (2010); BEC: Moore et al. (2013); TOPAZ2: Dunne et al. (2010, 2013); Diat-HadOCC: Palmer and Totterdell (2001); Martin et al. (2011), PELAGOS: Vichi et al. (2007, 2011); PISCES: Aumont and Bopp (2006); Lengaigne et al. (2009); Aumont et al. (2015), NPZD: Watanabe et al. (2011); Kawamiya et al. (2000); HAMOCC: Maier-Reimer et al. (2005); Ilyina et al. (2013). We only use the most recent peer-reviewed reference of each MBM. MBMs are only listed once, even though some of them are used in more than one ESM. The Roman numerals refer to the biologically-driven mechanisms while P, Z, C, cG, S, B denote the organism groups phytoplankton, zooplankton, calcifiers, coastal gas producers, surface mat producers and bacteria. Organism groups that are not explicitly described but parametrized are in parenthesis. Checkmarks with additions refer to the biogeochemical carbon pump (bcp), DMS or $N_2O$ (specific SCCs), light absorption by neutrally/negatively buoyant phytoplankton (LA-nbp) or to the numbers of explicitly described functional groups. In PISCES, the SCCs are not included by default but available through additional modules.

## 5 What needs to be done: An alternative way to design the marine biological component of ESMs

The mechanism diversity in today's Earth system models is low although the marine biological modules include a relatively large number of biological variables. In fact, most of the models include more functional groups than we think are necessary to capture all three classes of mechanisms. Hence, it should be relatively

5 easy to increase the mechanism diversity and the desired more complete description of links between marine biota and other Earth system components.

Given the current level of process understanding, we propose to keep the organic carbon and alkalinity pumps and add at least one gas shuttle, and light absorption. In parallel, pilot studies with biogenic primary aerosols should be conducted and sensitivity experiments with the other two biogeophysical mechanisms

10 should be performed. Further mechanisms may have to be added with improved process knowledge or increasing model resolution while others may have to be omitted, because they turn out to be negligible. So the list of mechanisms is not fixed.

To capture the suggested mechanisms in ESMs, only few additional functional groups are needed. Calcifiers and coastal gas and surface mat producers should be explicitly taken into account. Parametrizing

15 calcification may work out for today's ocean, but in climate change scenario experiments, this parametrization may no longer be appropriate. Under future acidified conditions, the composition of calcifying and non-calcifying species of the phytoplankton as well as the growth behavior of calcifiers may significantly change due to competing selection pressures. To allow for such shifts in community composition, calcifiers should be explicitly implemented as a separate state variable. Surface mat producers, represented by

cyanobacteria, are included in a few ESMs (e.g. Dunne et al., 2013), because of their role as nitrogen fixers in the nitrogen cycle. Their role in the biogeophysical mechanisms is not included and we suggest to account for that by adding the trait "surface buoyancy".

Our knowledge of other sensitivities is still underdeveloped, so it would be premature for example to include functional dependencies of the pH-effect on phytoplankton growth. The same is true for our knowledge about genetic adaptation towards climate stressors.

## 6   Summary and Conclusions

We distinguish three main classes of biologically-driven climate-relevant mechanisms. We argue that a fundamentally different kind of progress will be achieved if members of all classes of mechanisms are included in ESMs for climate projections. To resolve the mechanisms, five functional groups are needed, including bulk phyto- and zooplankton, calcifiers as well as coastal gas and surface mat producers. Thus, our suggested marine biosphere module for ESMs may be even less complex than those modules currently used for climate projections. But in contrast to these state-of-the-art concepts, a wider range of important links between the marine biosphere and other Earth system components – and consequently more feedbacks – is allowed.

We believe that mechanism diversity is better suited to account for possible changes in ocean biota and consequences for the climate system. With global warming and ocean acidification, the marine biota will be altered (e.g. Hallegraeff, 2010). Since key groups respond differently to climate change, the strength of biologically-driven mechanisms will also change, and consequently links to other Earth system components. The feedback loops associated with these mechanisms will be altered accordingly. Thus, to evaluate the response of the climate system, the mechanism diversity should be increased.

*Acknowledgements.* Numerous discussions with and valuable comments from Aike Beckmann are highly appreciated. Thanks also go to three anonymous reviewers and the Editor for helpful comments as well as to Christine Nam for proofreading. An earlier version of this manuscript has been commented by Dallas Murphy, Jochem Marotzke and the participants of the "Advanced Scientific Writing Course for Faculty" that took place at the Max Planck Institute

5 for Meteorology in Hamburg. IH has been financed through Cluster of Excellence "CliSAP" (EXC177), University of Hamburg, funded by the German Science Foundation (DFG). Part of this study was also funded by German BMBF project SOPRAN (Surface Ocean Processes in the Anthropocene), SOPRAN 03F0662E. SeS has been supported by the DFG-funded SPP 1689 "Climate Engineering".

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
