# Peer review of "Climate-Relevant Marine Biologically-Driven"

_Biogeosciences, 2016_

## Referee Comment (RC1) · Anonymous Referee #2 · 16 Aug 2016

The manuscript is a very nice review of the state of ocean biogeochemistry as implemented in current Earth System Models. (ESMs). It proposes that to better account for climate relevant mechanisms arising from element cycling in the ocean, three mechanisms (carbon pumps, gas and particle transfer, biophysical) should be included, which ideally could be represented by five plankton functional types. The manuscript is clearly organized, well written and addresses the currently much debated topic of climate-carbon cycle feedbacks. It is therefore of interest to a large community of climate scientists.

The state of current knowledge of the individual mechanisms (carbon pumps, gas and particle transfer, biophysical) is well documented from the recent literature. However,

to better convince the reader of the actual need to include more PFTs in ESMs, it would be nice to include a section on the sensitivities to the most common stressors (temperature, pH, oxygen).

Lines 79-81 state that marine calcifiers are needed as a functional group to correctly simulate alkalinity fields. This is a strong statement, which may need to be modified and/or explained in more detail. Model simulations including calcification as part of phytoplankton production have shown difficulties in accurate alkalinity representation, because of small biogeochemical effects compared to large circulation signals (e.g. Koeve et al. GMD 2014). Therefore, the expression 'correctly' is probably overstating what models are presently able to reproduce. Furthermore, the text reads as if one PFT would allow representing ocean calcification as a whole and (if at all) current models are including phytoplankton calcifiers based on coccolithophorids. However, the inclusion of aragonite producers (Gangstoe et al. BG 2008) showed that shallow calcite dissolution and thus alkalinity fields could be better simulated compared to pure consideration of coccolithophorids. Other calcifying organisms such as corals and foraminifera may play equally important roles in different ocean regions. Because of the very different organisms, probably contributing comparable amounts to global calcification, some more critical discussion if/how this could be solved by a single PFT would be desirable.

---

## Referee Comment (RC2) · Anonymous Referee #3 · 19 Aug 2016

I am sorry to say that I do not think the present contribution is suitable for inclusion in this special issue. It reads like a rough first draft of an "Ideas and Perspectives" paper, and needs a substantial rethink. Doing this will benefit the authors themselves as well as the broader Earth System Modelling community, but will take longer than is likely to be available for this issue.

The present draft tries to cover too much and ultimately achieves little. It is concerned mainly with special pleading for more consideration of a few processes in which the authors have invested time and developed expertise, but does not make a strong case for why these processes, and not others, should be given more attention by the ESM community.

[Figure]

The English is poor. If at all possible the authors should enlist a colleague entirely fluent in English (preferably a native speaker) to help refine the MS before resubmission. In such a commentary (vs a primary scientific contribution) attention to the details of language is particularly important. (Note also that terms like 'albedo' and 'radiative forcing' are used in a naive fashion. The latter in particular is often contentious and needs to be used (if at all) in a manner consistent with its usage in the existing literature.)

The paper considers the need both for additional biological processes and more focus on the coastal zones, but does not make a strong case for either. Ocean circulation is taken for granted and the technical challenges of resolving the nearshore in global-scale models are not considered. The spatial resolution required to resolve ocean circulation e.g. in the North Sea is such that running models at global scale with this resolution is simply not possible. Nesting, downscaling and adaptive grids are all approaches that can be used in complementary ways to fill information gaps, but there is no discussion in this commentary of the literature on these topics. Embedding in existing global scale circulation models models of biological processes that we know to be important in coastal zones achieves nothing (garbage in - garbage out).

I am reminded of the commentary of Prof. Myles Allen in Nature 425: 242 (2003), who stated that the "challenge of probabilistic - or risk-based - climate forecasting is to start saying what changes can be ruled out as unlikely, rather than simply ruled in as possible". The current contribution is not concerned with such forecasts, but I find this statement relevant and instructive. The manuscript offers up a shopping list of ocean biogeochemical processes that might be important for climate (ruled in as possible) but lacks clear direction in discussing which ones the authors think should be given priority. Their criticism of existing practice has a 'straw man' quality to it, e.g., on 1/13-15. Who exactly articulated such a strategy?

What are the criteria for a process to be considered globally important? In this essay there is no discussion of this question that could reasonably be described as quantitative. Processes may be locally important but average to zero globally; nonlinear
rectification effects may be demonstrable but still second order at global scale.

The choices for prioritization are unconvincing. N2O, for example, is more or less dismissed out of hand. The reasons given for dismissing it are incorrect (e.g., Dore et al. 1998 Nature 396: 63; Lueker et al 2003 10.1029/2002GL016615), and the literature that shows that it may be an important climate feedback (e.g., Jin and Gruber 2003 10.1029/2003GL018458) is not considered. DMS on the other hand is given pride of place as an important climatic driver, and more attention from Earth System Modellers is recommended, but recent literature suggesting that it is actually a second order effect (e.g., Quinn and Bates 2011 10.1038/nature10580) is ignored. The emphasis on the biophysical effects of changing ocean viscosity is quite perplexing. It may be true that this is an important climate driver that has been neglected. Or, more likely, it may prove to be an interesting (if rather esoteric) subject for research, but of negligible importance for climate. These authors make no effort to explain why they think it should be prioritized relative to the dozens of other possibilities.

One thing the authors could do is make a table of all of the processes they discuss, and rank their importance in terms of future model development by criteria that are clearly stated and applied consistently. This might lead them towards crafting a sound and credible contribution.

---

## Referee Comment (RC3) · Anonymous Referee #4 · 21 Aug 2016

This is an interesting illustrative paper that will be helpful in guiding the discussion on the amount of complexity required in Earth System Models. The authors have identified some major mechanisms that have implications for climate feedbacks in the climate system. I can understand the underlying rationale of their selection and I agree with the proposed mechanisms, although I would stress the fact that the processes that are being listed under each mechanism may not be exhaustive. Another point of relevance is the role played by coastal processes. The authors seem to suggest that coastal parametrizations of nutrient supply may be sufficient to capture the most relevant aspects of the identified feedback-related mechanisms (this is for instance implemented in PISCES, both for macro and micro-nutrients). However, the next generation of ESMs

is moving towards higher resolutions, which would imply a better resolution of coastal processes and benthic-pelagic interactions. This would probably enhance the vertical supply in coastal regions, requiring a continuous readjustment of the parameterizations. I wonder if this is what the authors are suggesting as the way forward. Overall, the manuscript is well written and easy to read. It certainly deserves publication though it requires some minor clarifications, changes and amendments, as detailed below

P2L4: " . . . without going into ..." This is a main explanation for the choice of the processes. I think it would deserve some more words, as for instance some feedbacks may not be known in general or not completely known to the authors.

P2L8-18: The initial sentence implies that the functional group approach may not be adequate for the description of the system. I cannot see the authors finding any substantial counter-argument against this approach in the rest of the manuscript. On the contrary, it seems like they are even suggesting a more lumped approach, as for instance with the use of bulk zooplankton and phytoplankton.

P3L1-4 The role of dissolved organic matter in the biogeochemical pump has been thoroughly demonstrated in the field (Hansell et al., 2009, 2012; Kim et al., 2011) and in ESMs (Patara et al., 2012; Letscher et al., 2015). Particulate organic is only one pathway of carbon cycling, and likely to be mostly reduced in the anticipated more stratified conditions of a high $CO_2$ ocean.

P3L9-10 There are actually two forms of calcium carbonate with different dissolution rates used by different organisms. If the emphasis is on alkalinity, then both forms of $CaCO_3$ (and the relative biological pathways) should be considered or at least indicated why neglected.

Table 2. There are some inaccuracies in this table. The authors are invited to check the work by Laufkotter et al. (2015, their Table 1 and 2) to get a complete overview of the various processes implemented in the current generation of ESMs. In particular, TOPAZ has parametrized zooplankton and PELAGOS is the only model with separate

bacterioplankton and dynamic DOM cycling (not included in zooplankton). PELAGOS was used in CMIP5 in the CMCC-CESM model.

P4L25-31 It is a bit dismissive to state that the regions of low oxygen concentration are confined to the shelf seas. It has been demonstrated that the open-ocean oxygen minimum zones (e.g. Stramma et al., 2008) are expanding and the biogeochemical cycles of these gases may be enhanced in a warmer ocean (Wright et al., 2012).

P6L1-3 The inclusion of cyanobacteria would imply the inclusion of another pathway in the nitrogen cycle because these organisms are N-fixers. This should be clearly indicated and the possible interactions with M2 should be provided.

P6L5-7 I would tend to disagree that the buoyancy status is one major discriminant in underwater light attenuation by phytoplankton. The current representation of light absorption in ESM is rather crude and since most of the models listed in Table 1 have now a variable Chl:C ratio, it may be necessary to consider the specific attenuation of chl in various functional groups as this may affect the size of the oligotrophic regions of the subtropical gyres

References

Dutkiewicz, S., A. E. Hickman, O. Jahn, W. W. Gregg, C. B. Mouw, and M. J. Follows (2015), Captur- ing optically important constituents and properties in a marine biogeo-chemical and ecosystem model, Biogeosciences, 12(14), 4447–4481, doi:10.5194/bg-12-4447-2015.

Hansell, D. A., Carlson, C. A., Repeta, D. J., and Schlitzer, R.( 2009) Dissolved organic matter in the ocean: A controversy stimulates new insights, Oceanography, 22, 202–211

Hansell, D. A., Carlson, C. A., and Schlitzer, R. ( 2012) Net removal of major marine dissolved organic carbon fractions in the subsurface ocean, Global Biogeochem. Cy., 26, GB1016, doi:10.1029/2011GB004069.

Kim, J. M., Lee, K., Shin, K., Yang, E. J., Engel, A., Karl, D. M., and Kim, H. C. ( 2011) Shifts in biogenic carbon flow from particulate to dissolved forms under high carbon dioxide and warm ocean conditions, Geophys. Res. Lett., 38, L08612, doi:10.1029/2011GL047346.

Laufkotter, C., M. Vogt, N. Gruber, M. Aita-Noguchi, O. Aumont, L. Bopp, E. Buitenhuis, S. C. Doney, J. Dunne, T. Hashioka, J. Hauck, T. Hirata, J. John, C. Le Quere, I. D. Lima, H. Nakano, R. Seferian, I. Totterdell, M. Vichi, and C. Volker (2015), Drivers and uncertainties of future global marine primary production in marine ecosystem models, Biogeosciences, 12(23), 6955–6984, doi:10.5194/bg-12-6955-2015.

Letscher, R. T., J. K. Moore, Y.-C. Teng, and F. Primeau (2015), Variable c : N : P sto-ichiometry of dissolved organic matter cycling in the Community Earth System Model, Biogeosciences, 12(1), 209–221, doi:10.5194/bg-12-209-2015.

Patara, L., M. Vichi, and S. Masina (2012), Impacts of natural and anthropogenic climate variations on North Pacific plankton in an Earth System Model, Ecol. Model., 244, 132–147.

Stramma, L., G. C. Johnson, J. Sprintall, and V. Mohrholz (2008), Expanding oxygen-minimum zones in the tropical oceans, Science, 320(5876), 655–658, doi:10.1126/science.1153847.

Wright, J. J., K. M. Konwar, and S. J. Hallam (2012), Microbial ecology of expanding oxygen minimum zones, Nat Rev Micro, 10(6), 381–394, doi:10.1038/nrmicro2778.

---

## Author Response (AR1)

**Response to Reviewers' Comments**

November 23, 2016

**1 Reviewer #1**

We thank the reviewer for his/her constructive and helpful comments. Below, please find our point-by-point response (in blue color).

- The state of current knowledge of the individual mechanisms (carbon pumps, gas and particle transfer, biophysical) is well documented from the recent literature. However, to better convince the reader of the actual need to include more PFTs in ESMs, it would be nice to include a section on the sensitivities to the most common stressors (temperature, pH, oxygen).

  We would like to stress that it is not our primary goal to include *more* PFTs but rather to account for those that are *most important* for the climate system. Furthermore, we suggest to strive for higher mechanism diversity rather than more PFTs. Concerning the sensitivity to climate stressors, we agree that these aspects are relevant and have thus included them in a new section "What are the future impacts" of the revised version of the manuscript.

- Lines 79-81 state that marine calcifiers are needed as a functional group to correctly simulate alkalinity fields. This is a strong statement, which may need to be modified and/or explained in more detail. Model simulations including calcification as part of phytoplankton production have shown difficulties in accurate alkalinity representation, because of small biogeochemical effects compared to large circulation signals (e.g. Koeve et al. GMD 2014). Therefore, the expression 'correctly' is probably overstating what models are presently able to reproduce. Furthermore, the text reads as if one PFT would allow representing ocean calcification as a whole and (if at all) current models are including phytoplankton calcifiers based on coccolithophorids. However, the inclusion of aragonite producers (Gangstoe et al. BG 2008) showed that shal- low calcite dissolution and thus alkalinity fields could be better simulated compared to pure consideration of coccolithophorids. Other calcifying organisms such as corals and foraminifera may play equally important roles in different ocean regions. Because of the very different organisms, probably contributing comparable

amounts to global calcification, some more critical discussion if/how this could be solved by a single PFT would be desirable.

We fully agree that calcite and aragonite and the key organisms involved in the alkalinity dynamics need to be distinguished *if* the focus lies on the marine carbon cycle. Among the calcifiers, coccolithophores, however, are the most important group and mainly responsible for the vertical gradient in alkalinity. Other calcifying organism groups have been shown to be regionally important or are indeed assumed to be highly relevant for aragonite but only marginally for climate dynamics. From a climate perspective, the gain to represent calcifiers by more than one key group might be relatively small unless regional ESMs are applied; we are not aware of any study showing the added value with respect to climate relevance. Most importantly, the vertical alkalinity gradient needs to be generated; the carbonate chemistry should be represented in ESMs. With one additional key group, the calcifiers, represented by coccolithophores these basic features of the alkalinity pump will be achieved. We have extended the discussion in the subsection "M1 - biogeochemical pumps" to clarify this issue in the revised version of the manuscript.

**2 Reviewer #2**

We thank this Reviewer for his/her comments – we are reassured now that it is necessary and high time to stimulate a discussion about a more biological perspective on Earth System modeling. Below, please find our point-by-point response (in blue color).

- The present draft tries to cover too much and ultimately achieves little. It is concerned mainly with special pleading for more consideration of a few processes in which the authors have invested time and developed expertise, but does not make a strong case for why these processes, and not others, should be given more attention by the ESM community.

    In our manuscript we clearly state that we are interested in the **biologically**-driven mechanisms that are relevant for the climate and involve feedbacks among the different Earth system components. We provide a general framework for including the most important of these mechanisms in ESMs. This framework can persist even if new biological climate-relevant mechanisms are discovered. We completely revised our introduction and substantiated our choice of mechanisms.

- The English is poor. If at all possible the authors should enlist a colleague entirely fluent in English (preferably a native speaker) to help refine the MS before resubmission. In such a commentary (vs a primary scientific contribution) attention to the details of language is particularly important. (Note also that terms like 'albedo' and 'radiative forcing' are used in a naive fashion. The latter in particular is often contentious and needs to be used (if at all) in a manner consistent with its usage in the existing literature.)

    We fully agree that a clear language is important in scientific communication. However, the reviewer does not specify where clarity is missing in the text. Concerning the terms "albedo" or "radiative forcing", we are well aware that these two terms are used differently in the different communities or studies (see e.g. Chung and Soden, Environ. Res. Lett 10(7) 2015). Yet, the reviewer does not elaborate in which way we use these two terms in a "naive fashion".

- The paper considers the need both for additional biological processes and more focus on the coastal zones, but does not make a strong case for either. Ocean circulation is taken for granted and the technical challenges of resolving the nearshore in global-scale models are not considered. The spatial resolution required to resolve ocean circulation e.g. in the North Sea is such that running models at global scale with this resolution is simply not possible. Nesting, downscaling and adaptive grids are all approaches that can be used in complementary ways to fill information gaps, but there is no discussion in this commentary of the literature on these topics. Embedding in existing global scale circulation models models of biological processes that

we know to be important in coastal zones achieves nothing (garbage in - garbage out).

We would like to stress that it is not our goal to discuss general deficiencies in ESMs like circulation patterns but rather to look at the representation of climate-relevant mechanisms induced by the marine biology. Consequently, we do not aim at reviewing or summarizing all issues related to the representation of coastal zones in global models and the approaches to overcome them. Instead, we present our view on future needs of ESMs with respect to the representation of marine biology relevant in climate simulations. Also, it is not our goal to address any technical issues.

- I am reminded of the commentary of Prof. Myles Allen in Nature 425: 242 (2003), who stated that the "challenge of probabilistic - or risk-based - climate forecasting is to start saying what changes can be ruled out as unlikely, rather than simply ruled in as possible". The current contribution is not concerned with such forecasts, but I find this statement relevant and instructive. The manuscript offers up a shopping list of ocean biogeochemical processes that might be important for climate (ruled in as possible) but lacks clear direction in discussing which ones the authors think should be given priority. Their criticism of existing practice has a 'straw man' quality to it, e.g., on 1/13-15. Who exactly articulated such a strategy?

It is unclear to which part of the manuscript the reviewer is referring to with this comment and where on page 1 (l.13-15) we criticise a strategy that has not been articulated. We agree with this statement by Allen (2003) but don't see the point. First of all, we provide a general framework with classes of mechanisms (not processes - and we clearly distinguish processes and mechanisms, see p.1, line 21ff. and p.2, line 20ff.) – this framework is not a shopping list. Second, we consider **biogeochemical and biogeophysical** and not only biogeochemical mechanisms (we clearly distinguish these 2 types, p.4, line 33ff.). Please also note, that this is a manuscript submitted to the "Ideas and Perspectives" category, where personal opinions are not a reason for rejection.

- What are the criteria for a process to be considered globally important? In this essay there is no discussion of this question that could reasonably be described as quantitative. Processes may be locally important but average to zero globally; nonlinear rectification effects may be demonstrable but still second order at global scale.

We agree that in general there are aspects that are locally or regionally important but turn out to be less important on a global scale. Yet, in our manuscript, first of all we focus on climate-relevant and not necessarily globally important aspects. Second, we describe for each class of mechanisms in which way the climate system is affected and provide quantitative or at least semi-quantitative information. Unfortunately,

the climate effects of the different mechanisms cannot be directly compared. This would require to implement the mechanisms and to carry out systematic model runs. However, it is out of the scope of this study since we aim at *communicating our view* on which biologically driven mechanisms are climate-relevant and need to be captured in ESMs.

- The choices for prioritization are unconvincing. N2O, for example, is more or less dismissed out of hand. The reasons given for dismissing it are incorrect (e.g., Dore et al. 1998 Nature 396: 63; Lueker et al 2003 10.1029/2002GL016615), and the literature that shows that it may be an important climate feedback (e.g., Jin and Gruber 2003 10.1029/2003GL018458) is not considered. DMS on the other hand is given pride of place as an important climatic driver, and more attention from Earth System Modellers is recommended, but recent literature suggesting that it is actually a second order effect (e.g., Quinn and Bates 2011 10.1038/nature10580) is ignored. The emphasis on the biophysical effects of changing ocean viscosity is quite perplexing. It may be true that this is an important climate driver that has been neglected. Or, more likely, it may prove to be an interesting (if rather esoteric) subject for research, but of negligible importance for climate. These authors make no effort to explain why they think it should be prioritized relative to the dozens of other possibilities.

First, we provide references of specific articles to emphasize the importance of the biogeophysical aspects and kindly ask the reviewer to read them. Second, we would be very interested in a few examples for "the dozens of other possibilities". Third, we would like to stress that the framework, including the three classes of mechanisms, will persist even if new mechanisms are discovered or turn out to be climate-relevant. We stress this aspect now more clearly in our revised version. Fourth, concerning N2O, there is ongoing discussion about the climate relevance of marine sources. We admit that such discussions also have taken place with respect to DMS and added this aspect in our revised version. To be consistent, we also now consider the mechanism of N2O-production.

- One thing the authors could do is make a table of all of the processes they discuss, and rank their importance in terms of future model development by criteria that are clearly stated and applied consistently. This might lead them towards crafting a sound and credible contribution.

We argue that it is important to consider **mechanism diversity** and thus to rank the classes of mechanisms would be completely in contrast to our idea. In the introduction of our revised version, we formulated our idea more clearly. Nevertheless, we agree that it is helpful to describe what is needed most in near future. Thus we introduced a new section "What needs to be done".

**3 Reviewer #3**

We thank the reviewer for his/her constructive and helpful comments. Below, please find our point-by-point response (in blue color).

- I can understand the underlying rationale of their selection and I agree with the proposed mechanisms, although I would stress the fact that the processes that are being listed under each mechanism may not be exhaustive.

  We agree that the processes listed under each mechanism are not complete. Depending on the level of detail in the description of the mechanisms, the number of processes varies. We hope that we found a consistent way to include the most contributing processes for each mechanism. In addition, we completely revised our introduction and explain the rationale behind our choices of mechanisms.

- Another point of relevance is the role played by coastal processes. The authors seem to suggest that coastal parametrizations of nutrient supply may be sufficient to capture the most relevant aspects of the identified feedback-related mechanisms (this is for instance implemented in PISCES, both for macro and micro-nutrients). However, the next generation of ESMs is moving towards higher resolutions, which would imply a better resolution of coastal processes and benthic-pelagic interactions. This would probably enhance the vertical supply in coastal regions, requiring a continuous readjustment of the parametrizations. I wonder if this is what the authors are suggesting as the way forward.

  We completely agree that other biologically-driven mechanisms may become important, too, if for instance the spatial model resolution is increased. However, we are confident that the general framework with *classes of mechanisms* will persist even if it turns out that more *individual mechanisms* need to be included. We have addressed this issue in a new section "What needs to be done" in the revised version. Concerning technical issues, we have no easy solution to that, since they are not our focus. One approach could be to adjust the paramterizations to the respective resolution.

- ": without going into ..." This is a main explanation for the choice of the processes. I think it would deserve some more words, as for instance some feedbacks may not be known in general or not completely known to the authors.

  We completely revised our introduction section and motivate our choice of mechanisms more clearly.

- The initial sentence implies that the functional group approach may not be adequate for the description of the system. I cannot see the authors finding any substantial

counter-argument against this approach in the rest of the manuscript. On the contrary, it seems like they are even suggesting a more lumped approach, as for instance with the use of bulk zooplankton and phytoplankton.

It was not our attempt to criticize that PFTs are introduced. We only regret that new PFTs are included based on their role in biogeochemical cycles, specifically, the carbon cycle alone. We rephrased this part to make sure that this is not misunderstood.

- The role of dissolved organic matter in the biogeochemical pump has been thoroughly demonstrated in the field (Hansell et al., 2009, 2012; Kim et al., 2011) and in ESMs (Patara et al., 2012; Letscher et al., 2015). Particulate organic is only one pathway of carbon cycling, and likely to be mostly reduced in the anticipated more stratified conditions of a high CO2 ocean.

We assume that the reviewer is referring to the paragraph where we describe the classical biological carbon pump and sinking of organic matter. We agree that the *microbial carbon pump* is another way to sequester carbon over long time periods. However, to the best of our knowledge, there is no evaluation of the relevance of this pump for contemporary climate change. Even a large pool, like the marine RDOM, will have little impact on the climate system on time scales of some hundreds of years unless an imbalance between sources and sinks evolves. However, there are no indications or estimates for that, yet. We thus decided to only discuss this aspect in our revised version of the manuscript without proposing this mechanism to include into ESMs.

- There are actually two forms of calcium carbonate with different dissolution rates used by different organisms. If the emphasis is on alkalinity, then both forms of CaCO3 (and the relative biological pathways) should be considered or at least indicated why neglected.

We fully agree that calcite and aragonite and the key organisms involved in the alkalinity dynamics need to be distinguished *if* the focus lies on the marine carbon cycle, as also pointed out in the reply to reviewer #1. Among the calcifiers, coccolithophores, however, are the most important group and mainly responsible for the vertical gradient in alkalinity. Other calcifying organism groups have been shown to be regionally important or are indeed assumed to be highly relevant for aragonite but only marginally for climate dynamics. From a climate perspective, the gain to represent calcifiers by more than one key group might be relatively small unless regional ESMs are applied; we are not aware of any study showing the added value with respect to climate relevance. Most importantly, the vertical alkalinity gradient needs to be generated; the carbonate chemistry should be represented in ESMs.

With one additional key group, the calcifiers, represented by coccolithophores these basic features of the alkalinity pump will be achieved. We extended the discussion to clarify this issue in our revised version of the manuscript.

- There are some inaccuracies in this table. The authors are invited to check the work by Laufkotter et al. (2015, their Table 1 and 2) to get a complete overview of the various processes implemented in the current generation of ESMs. In particular, TOPAZ has parametrized zooplankton and PELAGOS is the only model with separate bacterioplankton and dynamic DOM cycling (not included in zooplankton). PELAGOS was used in CMIP5 in the CMCC-CESM model.

  Apparently, our table is not very clear. For PELAGOS, we lumped bacterio- and zooplankton together to avoid introducing an extra column - that is why we added a footnote. We have again checked all the entries and better arranged them in our revised version of the table.

- It is a bit dismissive to state that the regions of low oxygen concentration are confined to the shelf seas. It has been demonstrated that the open-ocean oxygen minimum zones (e.g. Stramma et al., 2008) are expanding and the biogeochemical cycles of these gases may be enhanced in a warmer ocean (Wright et al., 2012).

  We have completely revised this part and separately discussed the role of the marine biota in N2O-production and other greenhouse gases.

- The inclusion of cyanobacteria would imply the inclusion of another pathway in the nitrogen cycle because these organisms are N-fixers. This should be clearly indicated and the possible interactions with M2 should be provided.

  That is true, also other organism groups are involved in mechanisms relevant for the Earth system which indirectly may affect again the climate. But here we clearly want to concentrate on more immediate effects, those which alter the global energy (heat) content and distribution or change the content and distribution of greenhouse gases or ocean's turbulent viscosity. Otherwise it is inconsistent with the other mechanisms or the list of things that need to be considered will become too long. We explained our focus more clearly in the new introduction section.

- I would tend to disagree that the buoyancy status is one major discriminant in underwater light attenuation by phytoplankton. The current representation of light absorption in ESM is rather crude and since most of the models listed in Table 1 have now a variable Chl:C ratio, it may be necessary to consider the specific attenuation of chl in various functional groups as this may affect the size of the oligotrophic regions of the subtropical gyres.

A variable C:Chl-ratio is certainly a big step but insufficient keeping in mind that surface buoyant cyanobacteria (and other surface buoyant organisms) can build up condensed surface mats in the upper 1-2 m with consequences for the heat distribution. This feature cannot be reproduced by only taking into account a variable C:Chl-ratio.

**4    Editor's Comments**

We thank the Editor for his comments and the "translation" of some of the reviewers' comments. Below, please find our point-by-point response (in blue color).

- The last point of reviewer#3 is a serious one and should be considered in any case: "One thing the authors could do is make a table of all of the processes they discuss, and rank their importance in terms of future model development by criteria that are clearly stated and applied consistently. This might lead them towards crafting a sound and credible contribution." This cannot be easily dismissed. I think the present manuscript can benefit substantially from what reviewer#3 writes. This would also protect you as authors from potential similar critics of others. The more modest critics of reviewer#4 go somewhat into the same direction: ": without going into ..." This is a main explanation for the choice of the processes. I think it would deserve some more words, as for instance some feedbacks may not be known in general or not completely known to the authors." I think this refers to your sentence (p 2 l 3-4): "Instead of improving a single mechanism, the idea is to account for "mechanism diversity", without going into too much detail of one specific process at the expense of others." This is so general a statement that it is as such not very useful. The manuscript must become more concrete here. Rephrasing the one sentence will not be enough.

  We completely revised our introduction to explain our idea more clearly and also to motivate our choice of mechanisms. Instead of providing a table, however, we introduced a section "What needs to be done", where we concretely describe what next steps are doable.

- Concerning the alkalinity and CaCO3 production issue mentioned by reviewer#2, the effect of dilution with freshwater or the concentrating effect of evaporation should be mentioned as well. CaCO3 production/dissolution variations may not lead to large climate feedbacks on scenario time scales, but can be critical for natural variations of the carbon cycle and climate and also on feedbacks on the long-term. Therefore - not only for this CaCO3 related example - , it is crucial to define over which period and for which problem area the model in question should be applied.

  We introduced a new section "What are the future impacts" where we address all these issues.

- The summary and conclusion section is too concise in my view. How would look your "dream Earth system model" from a marine biologists view look like concretely? What would you suggest as next steps? The present statements such as the following one need to be substantiated by a concrete reasoning: "But in contrast to these state-of-the-art concepts, we consider important links between the marine biosphere

and other Earth system components and allow in that way more important feedback loops to take place. We believe that our framework is better suited to account for possible changes in the strength of feed backs."

We like keep the summary and conclusion section concise but we introduced a new section "What needs to be done", where we address these issues.

- I suggest that you provide a substantially revised manuscript based on all three reviews. Please, take all points of the referees thoroughly into account for this revision.

  We took all points of the reviewers very seriously and have substantially revised our manuscript; we would like to stress, however, that our focus lies on **biologically** driven mechanisms

---

## Author Response (AR2)

**Response to the Editor's Comments**

December 21, 2016

**1  Editior's Comments**

We thank the Editor for his comments. Below, please find our point-by-point response (in blue color).

- 1. Abstract: Replace "...enables a larger number of relevant feedbacks to take place..." by "...enables inclusion of the relevant feedbacks in climate change quantifications..."

  We slightly rephrased this sentence but would like to stress that it is too early to talk about "climate change quantifications".

- 2. The paper may cause different reactions to the readers depending on whether the read the paper with the eyes of a (more physical) climate modeller or with the eyes of a marine ecosystem modeller (see the spread in the reviews). Therefore, it would be good to state up-front what this paper is about. Please, insert a sentence directly after the sub-heading "1. Introduction": "This perspectives paper deals with the marine ecosystem component of Earth system models and the quantification of respective feedbacks as well as impacts." (or something along these words)

  We introduced a new opening sentence.

- 3. At end of page 1/beginning of page 1 you write: "To consider more biological variables in models is clearly the right way forward if one is interested in the response of the marine ecosystem to climate change (see Le Quéré, 2006, for a discussion)." Can you add reasons for this? I think this issue is debated. More complex models do not necessarily provide better results, especially when many not well constrained processes are added to models. I think one needs the appropriate level of complexity tailored to a specific purpose. Ernst's outstanding strength was to discriminate between important and unimportant processes in his modelling. Perhaps, you can add a paragraph discussing the trade-off between complexity and understanding? What

use is a model that is so complex that cause-effect links cannot be anymore identified?

We agree that this issue is under debate and thus changed the text accordingly.

- 4. Page 2, line 8: Replace "...trigger climate feedbacks..." by "...cause feedbacks to climate..."

Our understanding of a feedback is that it goes from A to B and back (and not just from A to B). We therefore just replaced "trigger" by "cause".

- "Bacteria decompose the organic matter while sinking down and thereby determine the efficiency of the organic carbon pump." This reads as if the bacteria are sinking down; moreover bacteria decompose also non-sinking organic matter. Please, reformulate so that the meaning is unambiguous.

We rephrased the sentence.

- 6. Page 3, line 9: "The climate relevance of the organic carbon pump has been evaluated by quantifying the air-sea fluxes of CO2 without this mechanism." This sentence does not make sense. I know what you mean, but for the unexperienced reader this is enigmatic. Please, reformulate.

We agree and rephrased the sentence.

- 7. Page 5, line 27: "Another important long-lived greenhouse gas is N2O with a global warming potential for a 100 year time horizon that is approximately 300 times higher compared to CO2 (Ramaswamy et al., 2001)." This cannot be. Or I have misunderstood what you write. The specific greenhouse potential (per molecule) of N2O is larger than that of CO2, but CO2 is and will be by far the largest agent in radiative forcing. Please, correct and use the newest results from IPCC AR5.

That CO2 is the strongest agent becomes clear when taking into account the actual atmospheric content of the GHG that is 3 orders of magnitude higher for CO2 than for N2O. However, we refer to GWP. We mentioned the original paper Ramaswamy et al., 2001 that was cited by the AR2. The new AR5-report gives the same number (295) but we agree that the newest reference should be taken, here and thus changed the reference accordingly.

- 8. Page 5, line 35: It would be good to mention potential destabilization of gas hydrates as well as sub-seabed permafrost and the conversion of the respective CH4 to CO2 by microbes.

While these processes may well be relevant for climate, we do not think that this is the right forum, because these pools are not related to **living** organic matter

(biota). The microbial conversion of CH4 to CO2 is a respiration pathway which is negligible compared to oxygenic respiration and will continue to be so in future (the release of CH4/CO2 by melting or destabilization may become climate relevant but the **biologically**-driven part plays a minor role). Thus, we chose not to add these processes here.

- 9. Page 6, sub-heading "3 What are the future impacts" You motivate the paper with quantification of climate relevant feedbacks. Therefore, the "impact topic" now comes a bit as a surprise. I do not know whether "impacts" is the best word here, as it is often contrasted to feedbacks. Maybe "What are the future implications for ecosystems and climate change?" is a more correct heading.

  We agree and changed the sub-heading.

- 10. Page 8, line 20: "Nevertheless, it is expected that an expansion of oxygen minimum zones will increase the production of N2O (Naqvi et al., 2010), because the highest production rates occur at the anoxic-oxic boundary layer." Nitrification is named as the key marine N2O production mechanism (Freying et al., 2012, Phil. Trans. R. Soc. B (2012) 367, 1245–1255). Boundary layers are usually associated with the friction layers at the top and bottom of the sea. It is not clear what is meant with "the highest production rates" – production rates of what? This sentence needs reformulation and correction. It may be wise to split the text into several sentences. The statements need backing by references.

  We rephrased the sentence, so that it is hopefully more clear now.

- 11. Please, give the manuscript to a native speaker for a language check and respective improvements. The respective comment by reviewer 2 ("The English is poor...") is not very diplomatically formulated (I know this hurts). Nevertheless this reviewer has a point and this needs to be addressed carefully.

  We have given the manuscript to Dr. Christine Nam, who is a native English speaker and a guest scientist at MPI-M, to check for language and have incorporated her suggestions in the manuscript.